# The Influence of Body Composition on the Systemic Exposure of Paclitaxel in Esophageal Cancer Patients

**DOI:** 10.3390/ph14010047

**Published:** 2021-01-09

**Authors:** Leni van Doorn, Marie-Rose B. S. Crombag, Hánah N. Rier, Jeroen L. A. van Vugt, Charlotte van Kesteren, Sander Bins, Ron H. J. Mathijssen, Mark-David Levin, Stijn L. W. Koolen

**Affiliations:** 1Department of Medical Oncology, Erasmus MC Cancer Institute, 3015 GD Rotterdam, The Netherlands; s.bins@erasmusmc.nl (S.B.); a.mathijssen@erasmusmc.nl (R.H.J.M.); s.koolen@erasmusmc.nl (S.L.W.K.); 2Department of Hospital Pharmacy, Erasmus MC, University Medical Center Rotterdam, 3015 GD Rotterdam, The Netherlands; m.crombag@erasmusmc.nl; 3Department of Internal Medicine, Albert Schweitzer Hospital, 3318 AT Dordrecht, The Netherlands; h.rier@erasmusmc.nl (H.N.R.); m-d.levin@asz.nl (M.-D.L.); 4Department of Surgery, Erasmus MC University Medical Center Rotterdam, 3015 GD Rotterdam, The Netherlands; j.l.a.vanvugt@erasmusmc.nl; 5Department of Pharmacy, Albert Schweitzer Hospital, 3318 AT Dordrecht, The Netherlands; C.vanKesteren@asz.nl

**Keywords:** body composition, paclitaxel pharmacokinetics, esophageal cancer

## Abstract

Changes in body composition are associated with chemotherapy-related toxicities and effectiveness of treatment. It is hypothesized that the pharmacokinetics (PK) of chemotherapeutics may depend on body composition. The effects of body composition on the variability of paclitaxel PK were studied in patients with esophageal cancer. Skeletal muscle index (SMI), visceral adipose tissue (VAT), and skeletal muscle density (SMD) were measured at the third lumbar vertebra on computed tomography (CT) scans performed before treatment. Paclitaxel PK data were collected from a prospective study performed between May 2004 and January 2014. Non-linear mixed-effects modeling was used to fit paclitaxel PK profiles and evaluate the covariates body surface area (BSA), SMI, VAT, and SMD using a significance threshold of *p* < 0.001. Paclitaxel was administered to 184 patients in a dose range of 50 to 175 mg/m^2^. Median BSA was 1.98 m^2^ (range of 1.4 to 2.8 m^2^). SMI, VAT, and SMD were not superior to BSA in predicting paclitaxel PK. The additive value of SMI, VAT, and SMD to BSA was also negligible. We did not find evidence that paclitaxel dosing could be further optimized by correcting for SMI, VAT, or SMD.

## 1. Introduction

Paclitaxel is a highly lipophilic antineoplastic agent and is administered as an intravenous infusion. It is widely used for the treatment of lung, ovarian, breast, and esophageal cancer, amongst others [1,2,3]. Paclitaxel is currently dosed solely based on the body surface area (BSA) of the patient.

Despite this BSA-individualized dose, the interindividual variability (IIV) of paclitaxel pharmacokinetics (PK) remains high and consequently, the variability in clinical outcome (i.e., efficacy and toxicity) remains high as well. Apparently, a part of the total IIV can be explained by BSA [4], which can be expected, as BSA is calculated from only height and weight as a surrogate for body composition. This may not take into account the actual differences affecting paclitaxel PK between patients [5,6,7]. Paclitaxel is poorly soluble in water and therefore the infusion fluid contains a micelle-forming agent, Cremophor EL^®^ [8]. The clearance of paclitaxel in this micelle-forming formulation is significantly increased in obese patients [9]. In addition, the time-above-threshold-concentration of 0.05 µmol/L is related to both hematological toxicities and peripheral neuropathy [1]. While a low paclitaxel clearance puts patients at risk for drug-related toxicities, patients with a high clearance are at risk of suboptimal systemic drug levels, leading to a diminished therapeutic effect. Ideally, other covariates—or sets of covariates—than BSA would be used to predict paclitaxel exposure before treatment initiation. Skeletal muscle mass (i.e., skeletal muscle index, SMI), adipose tissue, and skeletal muscle density (SMD) (i.e., a measure for skeletal muscle quality and intramuscular fat infiltration) could potentially serve as predictive covariates, as they are associated with altered volumes of distribution, metabolism, and clearance of cytotoxic drugs [10]. Previous studies demonstrated a wide variation in muscle mass and visceral adipose tissue (VAT) in patients with identical BSA and/or body mass index (BMI), producing a heterogeneity in chemotherapy tolerance and treatment-related toxicity such as neutropenia [6,11,12].

These findings suggest that SMI, VAT, and SMD may be superior to BSA or may add reliability in predicting drug exposure and could help optimize chemotherapy dosing strategies. More knowledge on SMI, VAT, and SMD influencing paclitaxel pharmacokinetics (PK) may therefore help to improve the individualization of paclitaxel dosing. Currently available population paclitaxel PK models lack actual bio-impedance measurements and merely apply different formulas using patients’ weight and height rather than specific metabolic parameters such as SMI, VAT, or SMD.

Patients with esophageal cancer are prone to common symptoms such as malnutrition and weight loss, which can lead to skeletal muscle wasting and loss of adipose tissue. These patients may show a higher IIV of paclitaxel PK and be at an increased risk of toxicity. In this study, we investigated whether variation between patients in paclitaxel exposure can be explained by metabolic parameters such as SMI, VAT, and SMD.

## 2. Results

In total, 550 paclitaxel plasma concentrations were available from 184 patients for PK analyses, as depicted in Table 1. Paclitaxel was administered intravenously to 147 males and 37 females. The median age in the total patient cohort was 64 years (range of 40 to 83 years). One hundred and thirty-two patients received paclitaxel 50 mg/m^2^ (72%), forty-five patients were treated with 100 mg/m^2^, and 7 patients with 175 mg/m^2^ paclitaxel.

The previously developed four-covariate model [13] including BSA, was able to fit the paclitaxel exposure data and showed plausible parameter estimates. However, a few parameters could not be well estimated at the moment of paclitaxel dosing for which the previously reported values [13] were used. This was the case for the peripheral distribution volume and the effect of total bilirubin on paclitaxel elimination. For the final BSA model, the “goodness-of-fit” data of observed versus predicted paclitaxel exposure and the visual predictive check (VPC) results are depicted in Figure 1 and Figure 2, respectively.

Replacing the covariate BSA by the actual bio-impedance measurements SMI, VAT, or SMD did not improve model fit (difference Objective Function Value (dOFV) +29, +34, and +25, respectively). Besides, the IIV of the elimination capacity of paclitaxel was increased (+4.3% for SMI, +5.5% for SMD, and +3.1% for VAT, respectively), as shown in Table 2. The influence of either BSA, SMI, VAT, or SMD on the estimated VM_EL_ of paclitaxel is depicted in Figure 3. In this model, BSA is positively correlated with the elimination capacity of paclitaxel (VM_EL_). For model specifications, see Section 4.4.

Furthermore, adding either covariate SMI, VAT, or SMD to the previously established covariate model including BSA did not reach our significance threshold of *p* < 0.001 (dOFV of −0.1, −0.1, and +3.4 respectively; data not shown). Data evaluation using a 2-compartmental model, in which no basic PK parameters needed to be fixed, led to similar results and did not alter our conclusion on the impact of BSA, SMI, VAT, and/or SMD (data not shown).

## 3. Discussion

To our knowledge this is the first study that assessed the direct correlation between PK of paclitaxel and the body composition parameters SMI, VAT, and SMD from cross-sectional CT images. Variation in paclitaxel exposure in relation to these body composition parameters was investigated. BSA was previously found to only have a clinically relevant impact on VM_EL_ [13,14,15]. Hence, we evaluated the influence of SMI, VAT, and SMD on VM_EL_. We found that the parameters SMI, VAT, and SMD did not give a significantly better model fit than BSA nor did they lead to a decrease in IIV of VM_EL_. Thus, these actual bio-impedance measurements were not superior to BSA in predicting paclitaxel PK. Moreover, the added value of these actual bio-impedance measurements to BSA also appeared negligible. Thus, the relatively high IIV of paclitaxel exposure could not be attributed to differences in SMI, VAT, or SMD. Therefore, according to our model, conventional BSA-based dosing of paclitaxel remains the best approach to dose paclitaxel and minimize paclitaxel IIV.

Recently, several studies suggested a correlation of SMI, VAT, and/or SMD with taxane-related toxicity. One example is a study in a cohort of 151 early breast cancer patients treated with anthracycline and docetaxel or paclitaxel in which patients with a low SMI had significantly more adverse events [16]. Another study correlated visceral adipose tissue with safety parameters in 1395 patients with non-metastatic breast cancer treated with an anthracycline and docetaxel and/or paclitaxel and found that patients with larger visceral adiposity had a lower cumulative dose suggesting a lower tolerability for the treatment [11]. These observations can be explained by the influence of adipose tissue on the taxane pharmacokinetic profile, and pharmacokinetics was correlated with body composition. However, pharmacokinetic data was lacking in these studies to support this hypothesis. Our findings indicate that actual bio-impedance measurements from CT scans cannot explain variability in paclitaxel PK.

A possible explanation for the low predictive value of SMI, VAT, and SMD in our study may be that the CT scans and the PK sampling were not performed on the same day. While BSA was always available on the actual PK day, the CT scan was performed before treatment initiation within 8–10 weeks. Another possible explanation is that the total number of PK samples is too small or the study population is too homogeneous to demonstrate the potential influence of these measured body size parameters as compared to BSA. In addition, our study has several limitations. It should be noted that our cohort consisted of patients with esophageal cancer and that the most of them (*n* = 132) were treated with the well-tolerable paclitaxel dosing schedule of 50 mg/m^2^ in a curative setting. Furthermore, not all blood samples were collected during the first treatment cycle resulting in different paclitaxel dosages, especially in the induction/palliative setting.

Since we cannot explain paclitaxel interindividual variability in pharmacokinetics, one may want to consider therapeutic drug monitoring (TDM). This has recently been extensively studied by Joerger et al. in a randomized controlled trial. Although paclitaxel TDM did not improve clinical outcome or severe neutropenia, it did improve tolerability in terms of paclitaxel associated neuropathy [17]. This extended cohort analysis in patients with esophageal cancer showed that SMI, VAT, and SMD were not superior to BSA in predicting paclitaxel pharmacokinetics. These parameters should therefore not be used for paclitaxel dosing. Our results do not support an alternative for BSA-based paclitaxel dosing.

## 4. Materials and Methods

### 4.1. Patients

The patient cohort comprised 184 adult patients with esophageal cancer treated with paclitaxel at the Erasmus MC Cancer Institute, who were prospectively included in an institutional database (www.trialregister.nl; NL2187 (NTR2311) between May 2004 and January 2014 [18,19]. All patients provided written informed consent for the mentioned trial, and only patients who received paclitaxel mono- or combination therapy were included. All patients with esophageal cancer received paclitaxel 50 mg/m^2^ weekly in a neoadjuvant chemoradiotherapy regimen [2], or as an induction or palliative treatment with paclitaxel 100 mg/m^2^ weekly for a maximum of 6 weeks, followed by a 175 mg/m^2^ dose every 3 weeks. From all patients, evaluable baseline computed tomography (CT) imaging of the abdomen was available.

### 4.2. Body Composition Measurements

BSA was calculated for each patient according the Mosteller method [20]. Body composition was assessed using each patient’s pretreatment staging CT scan prior to the start of paclitaxel treatment. The cross-sectional skeletal muscle surface area (SMA) and VAT were measured at the third lumbar vertebra (L3) level at one contrast-enhanced transversal CT-image slice and were automatically calculated using the preset Hounsfield Units (HU) thresholds and expressed in square centimeters using the in-house developed FatSeg software program package version 2.4 (developed by the Biomedical Imaging Group Rotterdam, Rotterdam, the Netherlands) using MeVisLab (Mevis Medical Solutions, Bremen, Germany). The measured SMA in cm^2^ was corrected for height squared (m^2^) to determine the skeletal muscle index (SMI; cm^2^/m^2^). SMD was quantified as mean muscle attenuation as assessed between −29 and +150 HU [21]. L3 was chosen as an anatomical landmark based on its linear correlation to total body lean body mass [22]. CT scans were performed within 8–10 weeks before treatment initiation. All CT scans were assessed on identical slices by a trained observer to whom patient details were blinded [23].

### 4.3. Paclitaxel Pharmacokinetics

The analyses for paclitaxel pharmacokinetics were performed according to previous studies [18,19]. According to protocol, three post-administration blood samples for PK analysis of paclitaxel were obtained up to 5 h after paclitaxel treatment using a formerly endorsed limited sampling strategy. The PK analysis was conducted in the first or in one of the following courses during one chemotherapy treatment cycle. Samples were collected in 4 mL lithium heparin (Li-He) blood collection tubes. Subsequent to sample collection, paclitaxel concentrations were quantitated by a validated high performance liquid chromatography/tandem mass spectrometry (HPLC/MS-MS) detection method [24]. Paclitaxel plasma concentrations below the lower limit of quantitation (LLOQ) of 2 ng/mL were not reported. Cremophor EL^®^, the formulation vehicle for paclitaxel, causes a shift in the blood distribution and reduces the availability of the free circulating fraction of paclitaxel. As a result, the total fraction of paclitaxel does not behave in a linear pharmacokinetic way in contrast to its free fraction.

### 4.4. Pharmacokinetic Model Evaluation and Covariate Analysis

A previously validated population PK model for paclitaxel was used as a reference model [15]. This three-compartment model with nonlinear elimination included four covariates: BSA, gender, age, and total bilirubin. These covariates were proven to significantly correlate with the elimination capacity of paclitaxel (VM_EL_). Firstly, we fitted the data to this model. Hereafter, we evaluated whether replacing BSA by other bio-impedance measures, including SMI, VAT, and SMD, improved the model fit, as depicted in Equation (1).
(1)$$VM_{EL}=\Theta_{1}*{\left(\frac{BI}{BI_{median}}\right)}^{\Theta_{2}}*\Theta_{3}{\;}^{GENDER}*{\left(\frac{AGE}{AGE_{median}}\right)}^{\Theta_{4}}*{\left(\frac{BILI}{BILI_{median}}\right)}^{\Theta_{5}}$$
where Θ_1_ represents the typical population value for maximal elimination rate of paclitaxel; *BI* represents the bio-impedance measurement BSA, SMI, VAT, or SMD; and Θ_2_ to Θ_5_ represent the estimated influence of the respective bio-impedance measurements, gender, age, and total bilirubin on the maximal elimination rate. Finally, we investigated the effect of adding either VAT, SMD, or SMI to the four-covariate model. All continuous covariates were centered to the population median value. Graphical diagnostics, differences in Objective Function Value (OFV) and IIV in *VM_EL_*, visual predictive check (VPC) with *n* = 1000, and parameter plausibility were used to evaluate whether actual bio-impedance measurements were superior or additive to the classical BSA approach. A significance threshold of *p* < 0.001, corresponding to a difference in OFV of >10.83 for one degree of freedom, was used to discriminate between the covariate models. Parameter precision was estimated using sampling importance resampling (SIR) [25].

Non-linear mixed effects modeling was conducted using NONMEM^®^ (version 7.3.0, ICON Development Solutions, Ellicott City, MD, USA) and Perl-speaks-NONMEM (version 4.4.8). All analyses were performed with the first-order conditional estimation method with interaction. Piraña^®^ (version 2.9.2) was used as interface and data management and graphical assessments were performed in R (version 3.0.1), e.g., using Xpose.

## 5. Conclusions

Our analysis support the current practice of dosing paclitaxel based on BSA. We did not find evidence that the paclitaxel dose can be further optimized by correcting for SMI, VAT and SMD.

## Figures and Tables

**Figure 1 pharmaceuticals-14-00047-f001:**
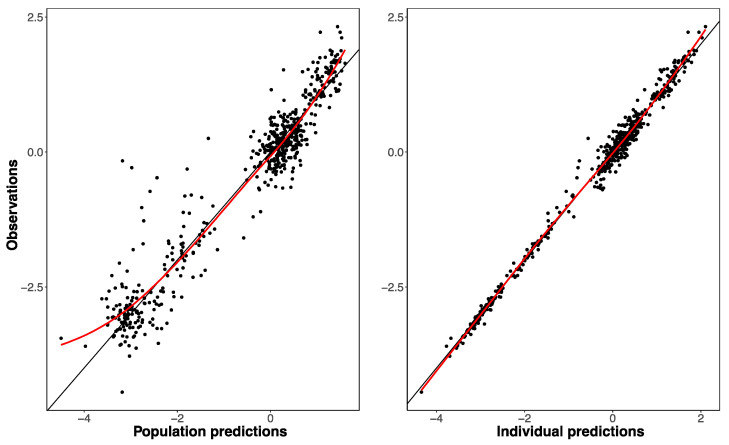
Goodness-of-fit plots presenting: BSA model predictions (**left panel**) or individual Bayesian predictions (**right panel**) versus observed paclitaxel concentrations, depicted using log transformed data.

**Figure 2 pharmaceuticals-14-00047-f002:**
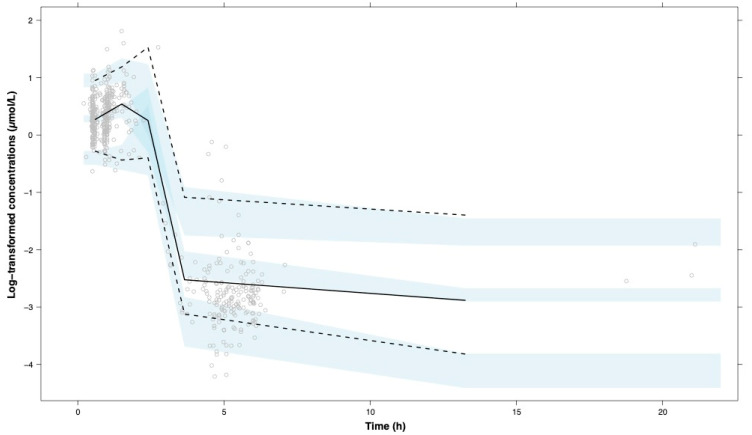
Visual predictive check plot of the BSA model using *n* = 1000 and log-transformed paclitaxel plasma concentrations. Dots represent observed paclitaxel concentrations, the black line represents the observed median concentrations, the dashed lines are the observed 5th and 95th percentiles, and the light blue areas represent the 95% confidence intervals of the median, 5th, and 95th percentiles.

**Figure 3 pharmaceuticals-14-00047-f003:**
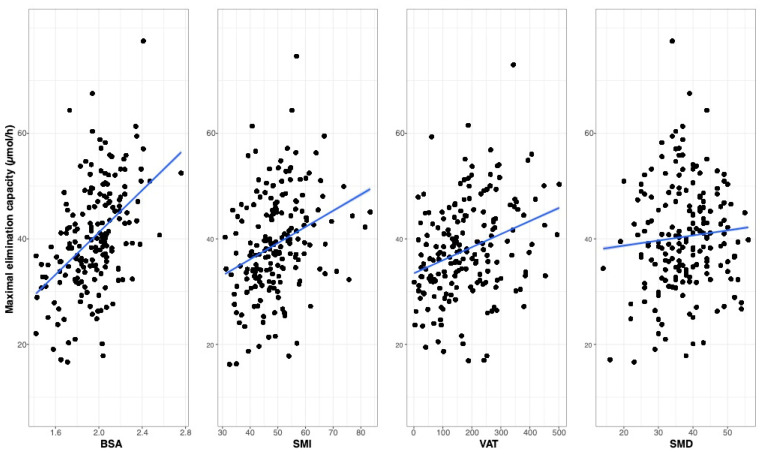
Data simulation of the impacts of BSA, SMI, VAT, and SMD on the maximal elimination capacity of paclitaxel.

**Table 1 pharmaceuticals-14-00047-t001:** Patient characteristics.

Parameters	Cohort
Number of patients (*n*)	184
Paclitaxel dose (mg/m^2^), median (range)	70 (62–252)
Infusion time (h), median (range)	0.9 (0.3–1.5)
Number of samples (*n*)	550
Per patient, median (range)	3 (2–4)
BSA (m^2^), median (range)	1.98 (1.42–2.76)
SMI (cm^2^/m^2^), median (range)	48.5 (30.9–83.4)
VAT (cm^2^), median (range)	165 (0.67–502)
SMD (HU), median (range)	37 (14–56)
Gender, male, *n* (%)	147 (80)
Age, median (range)	64 (40–83)
Indication, *n* (%) esophageal cancer	184 (100)
Paclitaxel treatment, *n* (%)	
Induction/palliative (3 weekly 175 mg/m^2^)	7 (4)
Induction/palliative (weekly 100 mg/m^2^)	45 (24)
Neoadjuvant (weekly 50 mg/m^2^)	132 (72)
Bilirubin, total (μmol/L), median (IQR)	7 (5–9)

BSA = body surface area, HU = Hounsfield Units, IQR = interquartile range, SMD = skeletal muscle density, SMI = skeletal muscle index, VAT = visceral adipose tissue.

**Table 2 pharmaceuticals-14-00047-t002:** Population pharmacokinetic parameters of paclitaxel of the covariate model of BSA, SMI, VAT, and SMD.

Parameter (Unit)	Covariate Model BSA		Covariate Model SMI		Covariate Model VAT		Covariate Model SMD	
	dOFV	REF	dOFV	+29	dOFV	+34	dOFV	+25
	Estimate	95% CI	Estimate	95% CI	Estimate	95% CI	Estimate	95% CI
VM_EL_ (µmol/h)	31.6	26.7–37.4	32.4	26.7–40.6 (11)	31.9	26.0–39.0	31.6	34.7–38.9
V_1_ (L)	24.0	20.8–27.6	24.7	21.3–28.4	25.0	21.8–28.4	25.2	22.0–28.8
V_3_ (L)	267	NA	267	NA	267	NA		
KM_EL_ (µmol/L)	0.40	0.29–0.52	0.51	0.37–0.73	0.49	0.33–0.68	0.55	0.36–0.74
VM_TR_ (µmol/h)	179	138–225	153	119–199	148	111–188	144	113–193
KM_TR_ (µmol/L)	1.91	1.37–2.67	1.80	1.46–1.98	1.75	1.24–2.39	1.73	1.15–2.36
K_21_ (h^−1^)	2.34	1.85–2.96	2.16	1.69–2.72	2.09	1.64–2.65	2.15	1.70–2.69
Q (L/h)	20.7	18.0–23.3	19.8	17.0–23.0	19.6	17.0–22.3	19.9	16.6–23.0
Body composition on VM_EL_	1.25	1.03–1.51	0.34	0.24–0.44	0.09	0.06–0.12	0.03	0.02–0.04
Age on VM_EL_	−0.30	−0.39–0.22	−0.30	−0.32–0.30	−0.50	−0.65–0.34	−0.28	−0.36–0.20
Gender on VM_EL_	1.07	0.96–1.19	1.20	1.06–1.34	1.23	1.09–1.37	1.30	1.15–1.47
Bilirubin on VM_EL_	−0.17	NA	−0.17	NA	−0.17	NA	−0.17	NA
*Interindividual variability*								
VM_EL_ (%)	24.3	20.8–28.3	28.6	24.9–33.3	27.4	24.0–32.4	29.8	25.5–34.9
V_1_ (%)	39.1	33.2–45.3	38.1	31.3–43.7	37.7	31.9–45.4	37.5	30.6–43.5
Q (%)	62.0	52.3–72.9	62.8	52.8–74.0	64.3	52.8–75.6	62.4	50.7–74.3
*Residual variability*								
σ_prop_ (%)	22.5	20.1–25.5	22.4	20.2–25.5	22.6	20.0–25.5	22.4	19.8–25.3

The data represent the following: BSA = body surface area, dOFV = difference Objective Function Value, K_21_ = rate constant of the distribution from the first peripheral compartment to the central compartment, NA = not applicable, Q = intercompartmental clearance between the central and second peripheral compartment, σ_prop_ = proportional residual error, SMD = skeletal muscle density, SMI = skeletal muscle index, VAT = visceral adipose tissue, V_1_ = volume of central compartment, V_3_ = volume of the second peripheral compartment, VM_EL_ = maximal elimination rate, VM_TR_ = maximal transport rate from the central to the first peripheral compartment, KM_EL_ = plasma concentration at half VM_EL_, and KM_TR_ = plasma concentration at half VM_TR_.

## Data Availability

The data presented in the current study are available from the corresponding author upon reasonable request.

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
