# Peer review of "The Influence of Body Composition on the Systemic Exposure of Paclitaxel in Esophageal Cancer Patients"

_pharmaceuticals, 2021, doi:10.3390/ph14010047_

Round 1
Reviewer 1 Report
It is a very interesting paper on the treatment of oesophageal cancer however the paper should be reorganised by putting the methods before the results. The discussion should reflect the objective of the study. Also, clearer paragraph of a conclusion can be added to make the answer clear co readers
Author Response
The authors are grateful for the reviewers’ efforts in improving this paper. In response to the reviewers’ comments, the specific modifications are as follows:
Comment 1. “It is a very interesting paper on the treatment of oesophageal cancer however the paper should be reorganised by putting the methods before the results. The discussion should reflect the objective of the study. Also, clearer paragraph of a conclusion can be added to make the answer clear co readers”
Response: Thank you very much for your encouraging comments. We have followed the “Instructions for Authors” for preparing our manuscript.
Reviewer 2 Report
It is significant in that this study was conducted on a relatively large number of patients with esophageal cancer for a long period of time, but the methodology is not noble, the clinical significance and usefulness are not high, and the description of methods and results are not clear. For publishing this paper, several points should be revised.
1. Considering the PK and physiochemical properties of paclitaxel, bio-impedance measurements such as SMI are unlikely to impact on PK of paclitaxel. In addition, the measurement of MI, VAT, and SMD are not common in actual clinical environments. The reason why the authors choose these bio-impedance measurements as a topic of this study should be described more clearly.
2. Bio-impedance indicators such as SMI appear to be dependent variables of BSA. It should be clarified whether this has been evaluated
3. It is doubt whether the model is valid. For example, 95% CIs of V3 are not presented in Table 1 and the result of VPC also appears to overestimate the concentration in Figure 2.
4. It seems necessary to show the diagram of the overall PK model including covariates.
Author Response
The authors are grateful for the reviewers’ efforts in improving this paper. In response to the reviewers’ comments, the specific modifications are as follows:
Responses to Reviewer 2.
Comments “It is significant in that this study was conducted on a relatively large number of patients with esophageal cancer for a long period of time, but the methodology is not noble, the clinical significance and usefulness are not high, and the description of methods and results are not clear. For publishing this paper, several points should be revised.”
Comment 1. “Considering the PK and physiochemical properties of paclitaxel, bio-impedance measurements such as SMI are unlikely to impact on PK of paclitaxel. In addition, the measurement of MI, VAT, and SMD are not common in actual clinical environments. The reason why the authors choose these bio-impedance measurements as a topic of this study should be described more clearly.”
Response: We agree with the reviewer that these chosen bio-impedance measurements are not common in actual clinical settings. However, these covariates can easily be calculated based on CT-scans, which are most often available before start of treatment. In the introduction we described our hypothesis that these parameters might be superior to BSA dosing (lines 51-60 and references 6, 11, and 12).
Comment 2. “Bio-impedance indicators such as SMI appear to be dependent variables of BSA. It should be clarified whether this has been evaluated.”
Response: The different bio-impedance indicators are likely to be highly correlated. Therefore it is not rational to include them both in a PK model. However, after we found that none of the tested bio-impedance indicators (SMI, VAT, SMD) performed better than BSA, we included them in the validated paclitaxel model besides the BSA covariate. For none of the bio-impedance factors this resulted in an improvement of the objective function value (OFV). This was stated in the results section. See lines 117-121.
Comment 3. “ It is doubt whether the model is valid. For example, 95% CIs of V3 are not presented in Table 1 and the result of VPC also appears to overestimate the concentration in Figure 2.””
Response: A few parameters could not be estimated for which the previously reported values were used. This was mentioned in the results section and was the case for the peripheral distribution volume. We respectfully disagree with the reviewer concerning the overestimation of paclitaxel plasma concentration, as goodness of fit plots and the VPC plot were within acceptable ranges.
Comment 4. “ It seems necessary to show the diagram of the overall PK model including covariates.
Response: The model development was extensively discussed by Joerger et al. (Clin Pharmacokinet 2012,51,607-617.), Crombag et al. (Pharm Res 2019, 36, 33.); respectively references 13 and 15 of our manuscript). Therefore we have chosen not to depict the overall PK diagram in this paper.
Reviewer 3 Report
Paclitaxel is a drug with high interindividual pharmacokinetics variability, currently dosed solely based on the body surface area. The originality of the study was to evaluate the relationship between body composition parameters (skeletal muscle index, visceral adipose tissue and skeletal muscle density) and Paclitaxel pharmacokinetics (based on three post-administration blood samples).
The methodology is clear presented, the pharmacokinetic evaluation was based on previously validated population PK model. Maybe it might be specified if the analysis was based only on pharmacokinetic parameters of Paclitaxel after the first cure or also on pharmacokinetic parameters after few cycles of Paclitaxel therapy.
Author Response
The authors are grateful for the reviewers’ efforts in improving this paper. In response to the reviewers’ comments, the specific modifications are as follows:
Responses to Reviewer 3.
Comments 1. “Paclitaxel is a drug with high interindividual pharmacokinetics variability, currently dosed solely based on the body surface area. The originality of the study was to evaluate the relationship between body composition parameters (skeletal muscle index, visceral adipose tissue and skeletal muscle density) and Paclitaxel pharmacokinetics (based on three post-administration blood samples).The methodology is clear presented, the pharmacokinetic evaluation was based on previously validated population PK model.
Response: Thank you very much for your encouraging comments.
Comments 2. “Maybe it might be specified if the analysis was based only on pharmacokinetic parameters of Paclitaxel after the first cure or also on pharmacokinetic parameters after few cycles of Paclitaxel therapy.”
Revision: The PK analysis was based on a single paclitaxel cycle per patient. The samples were drawn during the first or any of the following cycles. This is more clearly stated now in lines 200-201 of the revised manuscript.
Reviewer 4 Report
In the research article entitled “The Influence of Body Composition on the Systemic Exposure of Paclitaxel in Esophageal Cancer Patients” Doorn and co-workers have carried a study on 184 patients suffering from Esophageal Cancer by administering paclitaxel with dose loading ranging of 50 to 175 mg/m2 and investigated whether variation between patients in paclitaxel exposure can be explained by metabolic parameters such as SMI, VAT and SMD. The authors concluded that the conventional BSA-based dosing of paclitaxel remains the best approach to dose paclitaxel and these parameters should not be considered for paclitaxel dosing. Although the outcome of this study is negative but informative. My decision goes in the favor of publication of this work in pharmaceuticals.